# An exploratory machine learning study on paediatric abdominal pain phenotyping and prediction

Kazuya Takahashi [1,2*], Michalina Lubiatowska[1], Huma Shehwana[3], James K. Ruffle[1,4], John A. Williams[5,6,7], Animesh Acharjee[5,6,7], Shuji Terai[2], Georgios V. Gkoutos[5,6,7], Humayoon Satti[3‡], Qasim Aziz[1‡]

1 Centre for Neuroscience and Trauma, Wingate Institute of Neurogastroenterology, Blizard Institute, Barts and The London School of Medicine and Dentistry, Queen Mary University of London, London, United Kingdom, 2 Division of Gastroenterology and Hepatology, Graduate School of Medical and Dental Sciences, Niigata University, Niigata, Japan, 3 Department of Biological Sciences, National University of Medical Sciences, Rawalpindi, Pakistan, 4 Queen Square Institute of Neurology, University College London, London, United Kingdom, 5 College of Medical and Dental Sciences, Institute of Cancer and Genomic Sciences, University of Birmingham, Birmingham, United Kingdom, 6 Health Data Research United Kingdom, Midlands Site, Birmingham, United Kingdom, 7 Centre for Health Data Science, Birmingham, United Kingdom

‡ H.Satti and Q.Aziz are joint senior authors.
* kazuya911@med.niigata-u.ac.jp

## Abstract

### Background

The exact mechanisms underlying paediatric abdominal pain (AP) remain unclear due to patient heterogeneity. This preliminary study aimed to identify AP phenotypes and develop predictive models to explore associated factors, with the goal of guiding future research.

### Methods

In 13,790 children from a large birth cohort, data on paediatric and maternal demographics and comorbidities were extracted from general practitioner records. Machine learning (ML) clustering was used to identify distinct AP phenotypes, and an ML-based predictive model was developed using demographics and clinical features.

### Results

1,274 children experienced AP (9.2%) (average age: 8.4 ± 1.1 years, male/female: 615/659), who clustered into three distinct phenotypes: Phenotype 1 with an allergic predisposition (n = 137), Phenotype 2 with maternal comorbidities (n = 676), and Phenotype 3 with minimal other comorbidities (n = 340). As the number of allergic diseases or maternal comorbidities increased, so did the frequency of AP, with 17.6% of children with ≥ 3 allergic diseases and 25.6% of children with ≥ 3 maternal

**Data availability statement:** Data cannot be shared publicly because the Born in Bradford cohort retains ownership and control of the data. We used this data under a data sharing agreement between Queen Mary University of London and the Born in Bradford cohort. This agreement was contracted by Bradford Teaching Hospitals NHS Foundation Trust and Queen Mary University of London (see attached data sharing agreement). The original study was approved by the Bradford Research Ethics Committee (see attached approval document). Data access requests can be directed to the Bradford Teaching Hospitals NHS Foundation Trust (email: borninbradford@bthft.nhs.uk). Alternatively, data management for this project is overseen by Dr. Dan Mason, Head of Research Data at the Born in Bradford cohort (email: Dan.Mason@bthft.nhs.uk), and on the Queen Mary University side by Dr. Humayoon Satti (email: humayoon.satti@numspak.edu.pk).

**Funding:** This study was supported by JA Niigata Kouseiren Grant (Niigata University School of Medicine) and JSPS Grants-in-Aid for Scientific Research (grant number: 22K16013). The funders had no role in study design, data collection and analysis, decision to publish, or preparation of the manuscript.

**Competing interests:** The authors have declared that no competing interests exist.

comorbidities. The predictive model demonstrated moderate performance in predicting paediatric AP (AUC 0.67), showing that a child's ethnicity, paediatric allergic diseases, and maternal comorbidities were key predictive factors. When stratified by ML-predicted probability, observed AP rates were 18.9% in the < 40% group, 44.8% in the 40–50% group, 60.6% in the 50–60% group, and 100.0% in the > 60% group.

## Conclusions

This study identified distinct AP phenotypes and key risk factors using ML. Furthermore, the predictive ML model enabled risk stratification for paediatric AP. These analyses provide valuable insights to guide future investigations into the mechanisms of AP and may facilitate research aimed at identifying targeted interventions to improve patient outcomes.

---

## 1. Introduction

Abdominal pain (AP) is one of the most common symptoms among children and adolescents, with prevalence rates across the USA and Europe ranging from 0.3% to 19.0% [1–3]. In primary care, children presenting with AP are diagnosed with functional or medically un-explained AP in 80% of the cases, while organic causes are considerably less frequent [4,5]. Although rarely life-threatening, AP is often refractory to treatment and associated with psychiatric comorbidities, such as anxiety and depressive disorders [6], significantly affecting health-related quality of life [7]. Therefore, early recognition and intervention are warranted.

Organic diseases, as well as early life events ranging from allergy status, psychological comorbidity, and parental factors, are proposed risk factors for paediatric AP [5,8–10]. Moreover, these diverse risk factors can interact, contributing to its complexity and heterogeneity. A more robust stratification with large patient cohorts is essential for understanding the aetiology of paediatric AP, plausibly disclosing novel insights into its underlying mechanisms.

A robust approach for identifying subgroups of patients with shared characteristics is data-driven clustering by unsupervised machine learning (ML) [11,12]. Any yielded subgroups may share an underlying mechanism associated with AP. Furthermore, supervised ML is considered a powerful tool for clinical outcome prediction [13], and it could aid clinicians in assessing the risk of AP development in early childhood [14]. Based on this background, we hypothesised that an ML algorithm could classify children with AP into distinct phenotypes based on common characteristics, thereby helping to unravel the complex underlying AP mechanisms. Additionally, we hypothesised that ML algorithms could predict the development of paediatric AP and help identify the key factors associated with it.

In this exploratory study, we comprehensively evaluated the risk factors of paediatric patients with AP in a large birth cohort, covering a broad spectrum of cases. Using unsupervised ML, we delineated phenotypes of paediatric AP using paediatric and maternal clinical data. Moreover, supervised ML models tasked to predict the

development of paediatric AP uncovered underlying factors linked to its frequency. These complementary methods may serve as a catalyst for future research.

## 2. Materials and Methods

### 2.1. Participants

Between 2007 and 2011, 12,453 pregnant women (recruited at 26–28 weeks) and 13,858 children were registered in the Born in Bradford (BiB) cohort. The detailed demographics of the whole cohort are described elsewhere [15]. Data for this study were accessed for research purposes between November 2020 and December 2024 under a data sharing agreement. All data were fully pseudonymised before being accessed by the research team. Written informed consent was obtained from mothers at the time of cohort enrolment. For this study, we included children who had linked general practitioner (GP) records and whose comorbidities could be identified using the systematized nomenclature of medicine clinical terms (SNOMED-CT), a standardized terminology system used in medicine and healthcare. The details of the study population are illustrated in Fig 1. Ethical approval was obtained from the Bradford Research Ethics Committee (Ref: 07/H1302/112). This study was conducted following the principles of the Declaration of Helsinki.

### 2.2. Definition of AP and frequency survey of paediatric and maternal comorbidities

AP was defined as the presence of one of the following diagnoses in SNOMED-CT: AP, abdominal wall pain, or generalized AP. The frequencies of comorbidities, such as gastrointestinal (GI), psychological, and allergic diseases (asthma, eczema, urticaria, or hay fever), as well as diseases causing somatic pain, were also retrieved using SNOMED-CT, since these are associated with both organic and functional AP [8,10,16–18]. Additionally, the comorbidities of the children's genetic mothers were extracted to identify any associations between the maternal comorbidities and paediatric AP. The

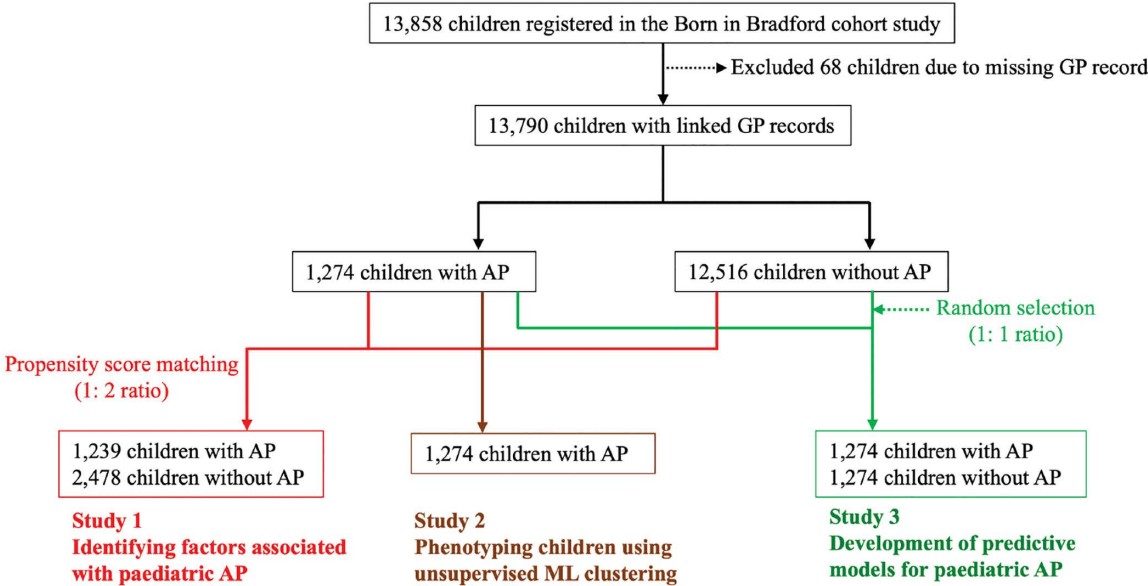

**Fig 1. Study population.** Among 13,858 children in the Born in Bradford cohort, 13,790 with linked general practitioner (GP) records were included. Study 1 identified factors associated with paediatric abdominal pain (AP) using propensity score matching (1:2 ratio). Study 2 performed phenotyping of AP using unsupervised machine learning (ML) clustering. Study 3 developed predictive models using supervised ML, including 1,274 children with AP and 1,274 randomly selected children without AP (1:1 ratio).

diseases of the children and their mothers investigated in this study and their SNOMED-CT codes are listed in S1 Table. Fathers were not included in our analysis pipeline due to a significant proportion of missing values.

### 2.3. Study 1. Identifying factors associated with paediatric AP

Study 1 aimed to determine the frequency of AP, elucidate the clinical characteristics of children with AP, and identify its associated factors. First, to adjust for background differences at the time of extraction between children with and without AP, we employed propensity score matching. After matching, we compared the two matched groups to assess differences in demographics and the frequency of comorbidities.

Next, we identified factors associated with paediatric AP. For this analysis, all comorbidities listed in S1 Table were included, irrespective of whether they occurred before or after the onset of AP, as the aim was to examine their overall association with paediatric AP.

### 2.4. Study 2. Phenotyping paediatric AP using unsupervised ML clustering

Study 2 focused on children experiencing AP, using unsupervised ML clustering to delineate phenotypes. The rationale for adopting unsupervised ML was to allow the algorithm to identify groups of children with similar characteristics from a large number of variables, minimizing human bias and thereby enabling the discovery of previously unrecognized phenotypes of paediatric AP. We considered this approach valuable as a potential foundation for future research.

In Python (version 3.7.12) [19], Uniform Manifold Approximation and Projection (UMAP), a non-linear dimensionality reduction technique, was applied to embed all variables into a three-dimensional latent space for subsequent clustering [20]. UMAP preserves both the local and global structure of high-dimensional data, placing data points with similar characteristics close to each other and thereby facilitating visual interpretation. A scatterplot was then generated from the latent features, and phenotypic AP clusters were identified using Hierarchical Density-based Spatial Clustering of Applications with Noise (HDBSCAN), a density-based hierarchical method [21]. HDBSCAN does not require a predefined number of clusters and is well suited for data with variable cluster densities. It produces a hierarchy of candidate clusters and selects the most stable ones, thereby yielding robust and reproducible clusters even in noisy datasets. The hyperparameters of UMAP and HDBSCAN used for clustering (settings that control how many clusters are found and how tightly the points are grouped) were summarized in S2 Table. Furthermore, to illustrate the hierarchical structure of the identified phenotypes, we adopted the condensed tree representation of HDBSCAN. This visualization allows the identification of cluster stability and the relative density distribution of each phenotype, providing insight into how clusters persist or split across different density thresholds [22].

Demographics and comorbidities of both children and their mothers were subsequently compared across the identified phenotypes to elucidate their distinct characteristics. Furthermore, we investigated whether the number of significant comorbidities identified in the phenotype analysis was associated with the frequency of paediatric AP.

### 2.5. Study 3. Using supervised ML to predict the development of paediatric AP

Supervised ML was used to predict paediatric AP. Supervised ML is useful for predicting the occurrence of clinical outcomes based on known labels — in this case, the presence or absence of AP — and has been widely applied for outcome prediction in previous studies [23,24]. Furthermore, by examining the variables used in the predictive model, clinically relevant factors associated with the outcome can be identified.

To prepare the data for model development, we constructed a balanced dataset by randomly selecting an equal number of children without AP from the control group to match those with AP (Fig 1), thereby mitigating potential model bias due to class imbalance. Predictive variables included demographic information and pre-existing comorbidities diagnosed prior to the onset of AP. Comorbidities with a prevalence of less than 1% were excluded to reduce the risk of model

overfitting. The variables used in the predictive model, along with their frequencies in children with and without AP, are summarised in S3 Table. The overall workflow for model development is illustrated in S1 Fig.

We initially developed predictive models for paediatric AP using several ML algorithms, including eXtreme Gradient Boosting, Categorical Boosting (CatBoost), Light Gradient Boosting Machine, and Random Forest. Model performance was assessed using receiver operating characteristic (ROC) curves and the corresponding area under the curve (AUC). In addition, accuracy, sensitivity, specificity, positive predictive value (PPV), and negative predictive value (NPV) were calculated. The model with the highest AUC was selected for subsequent feature importance analysis and risk stratification.

To interpret model predictions, we employed Shapley Additive exPlanations (SHAP) values. SHAP quantifies the contribution of each feature to individual predictions, allowing us to identify which variables had the greatest influence on the model's outputs and to enhance interpretability [25,26].

Finally, because the ML models produce continuous probability estimates for AP, we stratified children in the test dataset into four probability groups: <40%, 40–50%, 50–60%, and >60%. The observed risk of paediatric AP was then compared across these strata.

### 2.6 Statistical analysis

Continuous data were expressed as mean ± standard deviation (SD). Categorical data were expressed as numbers and percentages. The Student's t-test and Chi-squared test were used for numerical and categorical data, respectively.

To compare children with AP and those without, the propensity score was calculated using logistic regression analysis, with age at extraction and gender as covariates. Matched children with and without AP were then extracted. A 1:2 nearest-neighbour matching without replacement was performed, with a calliper width set to 0.2 times the standard deviation of the logit of the propensity score. Model discrimination was assessed using the c-statistic, and covariate balance was evaluated using standardized mean differences (SMDs), with SMD < 0.1 indicating good balance.

Univariate and multivariate logistic regression analyses were utilised to calculate odds ratios (ORs) and 95% confidence intervals (CI) for each variable, determining variables associated with paediatric AP. Variables significant in the univariate analysis were included in the multivariate analysis. As the comparisons between phenotypes were descriptive and intended to characterize group differences, Bonferroni correction was not applied. The Wald test was applied to compare ORs among the predicted probability groups. A p-value < 0.05 was considered statistically significant. EZR (Saitama Medical Center, Jichi Medical University, Saitama, Japan) was used for statistical analysis [27].

### 3. Results

#### 3.1. Study 1. Identifying factors associated with paediatric AP

*Summary: Paediatric AP was observed in 9.2% of children. Those with AP had more paediatric and maternal comorbidities. Key associated factors included Pakistani ethnicity, allergic and GI diseases, migraine, as well as maternal AP, allergic diseases, GORD, and migraine.*

A total of 13,790 children were included in the analysis, of whom 1,274 (9.2%) experienced AP (male/female = 615/659). The average age at diagnosis of AP was 5.6 ± 2.7 years. The characteristics of all children are summarised in S4 Table.

As a result of propensity score matching, 1,239 children with AP and 2,478 children without AP were selected (Table 1). Compared to children without AP, more children of Pakistani origin were in the AP group, while fewer were of White origin (all *p* < 0.01). Children with AP had higher rates of comorbidities, including allergic diseases, appendicitis, Celiac disease, constipation, and gastro-oesophageal reflux disease (GORD), and migraine (all *p* < 0.01). Moreover, mothers of children with AP reported higher incidences of AP, allergic diseases, arthritis, chronic muscle pain, functional dyspepsia, GORD, irritable bowel syndrome (IBS), and migraine (all *p* < 0.01).

**Table 1. Comparison of children with and without abdominal pain.**

| | Before matching (n = 13,790) | | | After matching (n = 3,717) | | |
|---|---|---|---|---|---|---|
| | Without AP (n = 12,516) | With AP (n = 1,274) | *p* value | Without AP (n = 2,478) | With AP (n = 1,239) | *p* value |
| Average age at the extraction ± SD (years) | 8.3 ± 1.1 | 8.4 ± 1.1 | <0.01 | 8.6 ± 1.1 | 8.6 ± 1.1 | 0.83 |
| Average age at the diagnosis of abdominal pain ± SD (years) | N/A | 5.6 ± 2.7 | N/A | N/A | 7.5 ± 2.3 | N/A |
| Gender, female, n (%) | 6,015 (48.1%) | 659 (51.7) | 0.01 | 1,304 (52.6) | 634 (51.2) | 0.40 |
| Route of birth, (vaginal), n (%) | 9,415 (77.0%) | 943 (75.3) | 0.17 | 1,866 (75.3) | 916 (73.9) | 0.36 |
| Ethnicity*, n (%) | | | | | | |
| Pakistani | 4,750 (45.8) | 703 (58.6) | <0.01 | 970 (46.9) | 681 (58.4) | <0.01 |
| White | 3,771 (36.4) | 267 (22.3) | <0.01 | 724 (35.0) | 259 (22.2) | <0.01 |
| Other ethnicities | 1,847 (17.8) | 229 (19.1) | 0.27 | 374 (18.1) | 227 (19.5) | 0.34 |
| Frequency of diseases, n (%) | | | | | | |
| Allergic diseases | 4,665 (37.3) | 601 (47.2) | <0.01 | 918 (37.0) | 584 (47.1) | <0.01 |
| Appendicitis | 24 (0.2) | 27 (2.1) | <0.01 | 4 (0.2) | 27 (2.2) | <0.01 |
| Arthritis | 6 (0.0) | 1 (0.1) | 0.49 | 0 (0.0) | 1 (0.1) | 0.16 |
| Celiac disease | 24 (0.2) | 9 (0.7) | <0.01 | 3 (0.1) | 9 (0.7) | <0.01 |
| Constipation | 142 (1.1) | 57 (4.5) | <0.01 | 30 (1.2) | 56 (4.5) | <0.01 |
| FD | 3 (0.0) | 4 (0.3) | <0.01 | 2 (0.1) | 4 (0.3) | 0.08 |
| GORD | 342 (2.7) | 66 (5.2) | <0.01 | 54 (2.2) | 63 (5.1) | <0.01 |
| IBD, colitis | 4 (0.0) | 2 (0.2) | 0.1 | 1 (0.0) | 2 (0.2) | 0.22 |
| Migraine | 53 (0.4) | 16 (1.3) | <0.01 | 10 (0.4) | 16 (1.3) | <0.01 |
| EDS, JHS | 9 (0.1) | 4 (0.3) | <0.01 | 1 (0.0) | 4 (0.3) | 0.03 |
| Autism | 36 (0.3) | 4 (0.3) | 0.87 | 9 (0.4) | 4 (0.3) | 0.84 |
| Intellectual disability | 16 (0.1) | 2 (0.2) | 0.78 | 3 (0.1) | 2 (0.2) | 0.75 |
| Mother's abdominal pain | 3,985 (31.8) | 631 (49.5) | <0.01 | 782 (31.6) | 616 (49.7) | <0.01 |
| Mother's allergic disease | 4,931 (39.4) | 609 (47.8) | <0.01 | 965 (38.9) | 597 (48.2) | <0.01 |
| Mother's appendicitis | 94 (0.8) | 13 (1.0) | 0.31 | 14 (0.6) | 13 (1.0) | 0.10 |
| Mother's arthritis | 495 (4.0) | 109 (8.6) | <0.01 | 24 (1.0) | 28 (2.3) | <0.01 |
| Mother's Celiac disease | 73 (0.6) | 14 (1.1) | 0.04 | 17 (0.7) | 13 (1.0) | 0.24 |
| Mother's chronic fatigue syndrome | 25 (0.2) | 3 (0.2) | 0.74 | 6 (0.2) | 3 (0.2) | 1.00 |
| Mother's chronic muscle pain | 167 (1.3) | 30 (2.4) | 0.01 | 30 (1.2) | 29 (2.3) | <0.01 |
| Mother's constipation | 128 (1.0) | 10 (0.8) | 0.55 | 36 (1.5) | 10 (0.8) | 0.09 |
| Mother's depressive disorder, bipolar disorder | 2,632 (21.0) | 309 (24.3) | 0.01 | 555 (22.4) | 300 (24.2) | 0.22 |
| Mother's FD | 162 (1.3) | 33 (2.6) | <0.01 | 36 (1.5) | 33 (2.7) | <0.01 |
| Mother's GORD | 876 (7.0) | 171 (13.4) | <0.01 | 172 (6.9) | 170 (13.7) | <0.01 |
| Mother's IBD | 84 (0.7) | 10 (0.8) | 0.59 | 12 (0.5) | 9 (0.7) | 0.35 |
| Mother's IBS | 877 (7.0) | 126 (9.9) | <0.01 | 171 (6.9) | 125 (10.1) | <0.01 |
| Mother's migraine | 1,845 (14.7) | 285 (22.4) | <0.01 | 378 (15.3) | 283 (22.8) | <0.01 |
| Mother's EDS, JHS | 47 (0.3) | 5 (0.4) | 0.93 | 10 (0.4) | 5 (0.4) | 1.00 |
| Mother's obsessive-compulsive disorder | 61 (0.5) | 10 (0.8) | 0.15 | 13 (0.5) | 10 (0.8) | 0.30 |
| Mother's schizophrenia | 6 (0.0) | 2 (0.2) | 0.17 | 1 (0.0) | 1 (0.1) | 0.62 |
| Mother's intellectual disability | 44 (0.4) | 1 (0.1) | 0.1 | 8 (0.3) | 1 (0.1) | 0.16 |

\* Gender and age at extraction were used to calculate the propensity score. Before matching, the standardized mean differences (SMD) for gender and age were 0.073 and 0.337, respectively. After matching, the SMD values for gender and age were reduced to 0.029 and 0.007, respectively, indicating that the data were well balanced. The c-statistic was 0.60 (95% CI: 0.58–0.61).

AP, abdominal pain; CI, confidence interval; SD, standard deviation; FD, functional dyspepsia; GORD, gastro-oesophageal reflux disease; IBD, inflammatory bowel disease; IBS, irritable bowel syndrome; EDS, Ehlers-Danlos syndrome; JHS, joint hypermobility syndrome.

Multivariate logistic regression analysis revealed that Pakistani ethnicity, allergic diseases, GI diseases, migraine, maternal AP, maternal allergic diseases, maternal GORD, and maternal migraine were significantly associated with paediatric AP (Table 2). Non–significant results are summarized in S5 Table.

### 3.2 Study 2. Phenotyping children using unsupervised ML clustering

*Summary: Unsupervised ML clustering identified three phenotypes of paediatric AP, characterized mainly by allergic diseases and maternal comorbidities. The risk of AP increased with the number of allergic diseases and maternal comorbidities, suggesting an additive effect.*

The unsupervised model classified children with AP into three distinct phenotypes: 137 children in Phenotype 1 (10.8%), 677 children in Phenotype 2 (53.1%), and 340 children in Phenotype 3 (26.7%) (Fig 2A). The remaining 120 children (9.4%) exhibited miscellaneous characteristics, making them unclassifiable by the model; these children were excluded from downstream analyses.

The condensed tree visualization (Fig 2B) revealed that Phenotype 1, located on the left, was smaller but highly stable, representing a dense and well-separated cluster. Phenotype 2, on the right, formed the largest and stable branch, suggesting a large, cohesive subgroup. Phenotype 3, positioned in the middle, appeared moderately dense and less stable, indicating a more diffuse structure. Overall, these hierarchical relationships underscore the robustness of the clustering process and the distinct density profiles among the identified phenotypes.

To further illustrate phenotypic characteristics, color-coded plots of representative pediatric and maternal comorbidities were generated (Fig 2C).

Subsequently, we compared the clinical characteristics among the three phenotypes (Table 3). Age at diagnosis of AP, gender, route of birth, and ethnicity showed almost no significant differences among the three phenotypes. The frequency

**Table 2. Significant results of logistic regression analysis for the diagnosis of paediatric abdominal pain.**

|  | Univariate | | Multivariate | |
|---|---|---|---|---|
|  | OR (95% CI) | *p* value | OR (95% CI) | *p* value |
| Pakistani (vs. other ethnicities) | 1.59 (1.37–1.83) | <0.01 | 1.58 (1.39–1.79) | <0.01 |
| Allergic diseases | 1.52 (1.32–1.74) | <0.01 | 1.19 (1.05–1.34) | 0.01 |
| Appendicitis | 13.8 (4.81–39.5) | <0.01 | 8.92 (4.94–16.1) | <0.01 |
| Celiac disease | 6.04 (1.63–22.3) | 0.01 | 3.35 (1.53–7.34) | <0.01 |
| Constipation | 3.86 (2.47–6.05) | <0.01 | 3.11 (2.23–4.33) | <0.01 |
| GORD | 2.40 (1.66–3.48) | <0.01 | 1.64 (1.23–2.18) | <0.01 |
| Migraine | 3.23 (1.46–7.14) | <0.01 | 2.95 (1.64–5.33) | <0.01 |
| Mother's abdominal pain | 2.14 (1.86–2.47) | <0.01 | 1.68 (1.48–1.90) | <0.01 |
| Mother's allergic diseases | 1.46 (1.27–1.67) | <0.01 | 1.17 (1.03–1.33) | 0.01 |
| Mother's arthritis | 2.36 (1.36–4.10) | <0.01 | 1.39 (0.89–2.16) | 0.14 |
| Mother's chronic fatigue syndrome | 1.96 (1.17–3.27) | 0.01 | 0.98 (0.28–3.41) | 0.98 |
| Mother's chronic muscle pain | 1.96 (1.17–3.27) | 0.01 | 1.12 (0.73–1.72) | 0.61 |
| Mother's FD | 1.86 (1.15–2.99) | 0.01 | 1.15 (0.76–1.74) | 0.51 |
| Mother's GORD | 2.13 (1.70–2.67) | <0.01 | 1.46 (1.21–1.76) | <0.01 |
| Mother's IBS | 1.51 (1.19–1.93) | <0.01 | 1.07 (0.86–1.33) | 0.55 |
| Mother's migraine | 1.64 (1.38–1.95) | <0.01 | 1.24 (1.06–1.44) | 0.01 |

FD, functional dyspepsia; GORD, gastro-oesophageal reflux disease; IBD, inflammatory bowel disease; IBS, irritable bowel syndrome; EDS, Ehlers-Danlos syndrome; JHS, joint hypermobility syndrome

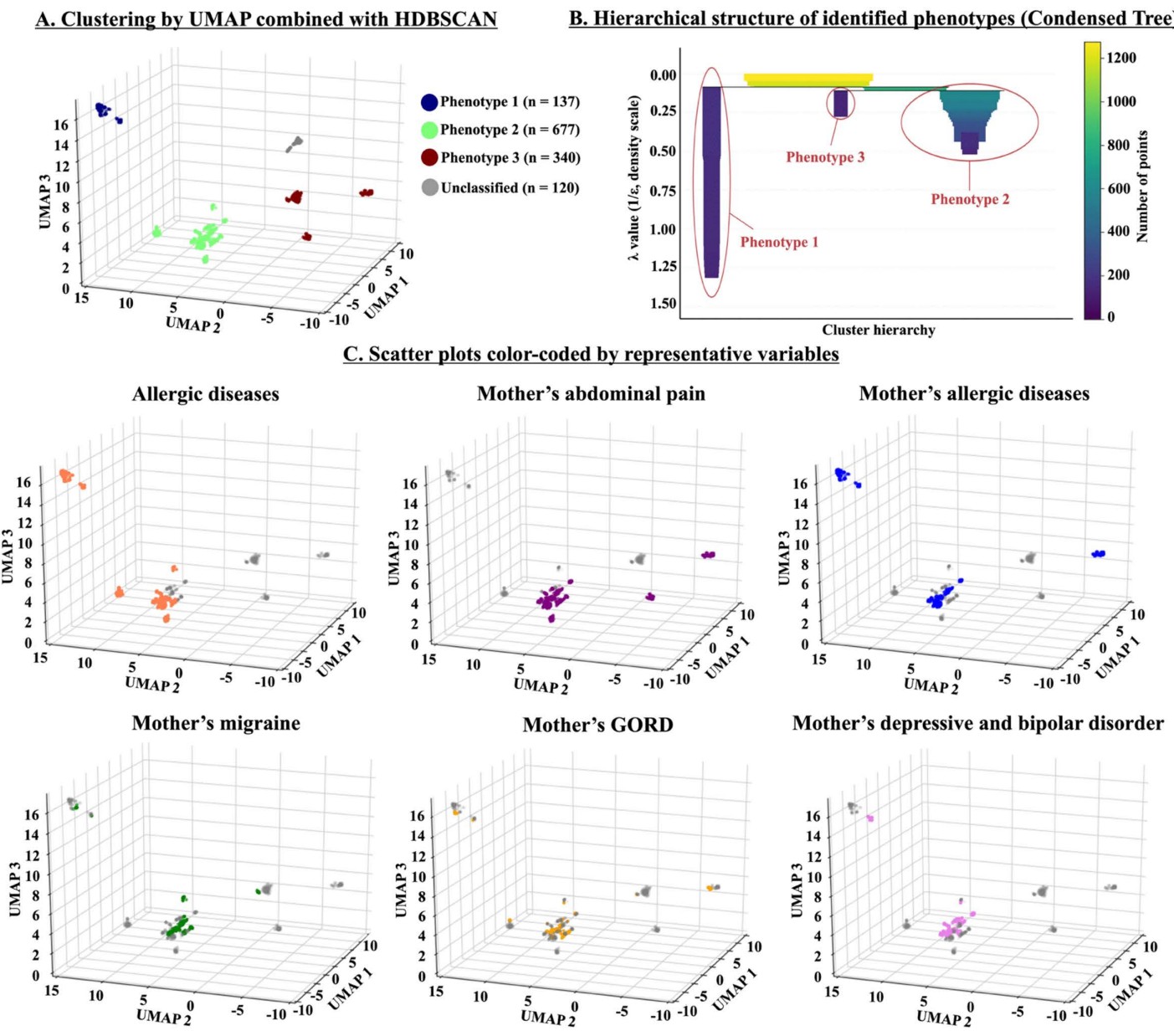

**Fig 2. Phenotyping of children with abdominal pain. A.** Scatter plots based on three new variables generated by unsupervised Uniform Manifold Approximation and Projection (UMAP). Hierarchical Density-Based Spatial Clustering of Applications with Noise (HDBSCAN) found 3 distinct phenotypes. **B.** Hierarchical structure of identified phenotypes (Condensed Tree). The vertical axis (λ value = 1/ε, density scale) represents the density scale used by the HDBSCAN algorithm. HDBSCAN performs hierarchical clustering by varying the distance threshold that defines neighborhood connectivity (ε). Smaller ε (larger λ) corresponds to higher-density regions, while larger ε (smaller λ) represents broader, lower-density structures. Clusters that persist across a wider range of λ values (longer vertical branches) indicate more stable and well-defined groups. The horizontal axis shows the cluster hierarchy, corresponding to different cluster branches identified by HDBSCAN. Each branch represents a cluster that emerges and persists across density thresholds. The color of each branch indicates the number of data points (cluster size), as shown in the color bar on the right. **C.** Scatter plots color-coded by representative variables. Each plot is colored according to the representative variables that characterize the three phenotypes.

**Table 3. Comparison of clinical characteristics among 3 pain phenotypes.**

| | Phenotype 1 (n = 137) | Phenotype 2 (n = 677) | Phenotype 3 (n = 340) | P value (Phenotype 1 vs. 2) | P value (Phenotype 1 vs. 3) | P value (Phenotype 2 vs. 3) |
|---|---|---|---|---|---|---|
| Average age at the extraction ± SD (years) | 8.5 ± 1.1 | 8.7 ± 1.1 | 8.6 ± 1.2 | 0.15 | 0.52 | 0.34 |
| Average age at the diagnosis of abdominal pain ± SD (years) | 5.6 ± 2.8 | 5.6 ± 2.7 | 5.6 ± 2.7 | 0.95 | 0.99 | 0.91 |
| Gender, female, n (%) | 67 (48.9) | 334 (49.3) | 172 (50.6) | 0.93 | 0.74 | 0.71 |
| Route of birth, (vaginal), n (%) | 111 (81.0) | 477 (70.5) | 260 (76.5) | 0.01 | 0.28 | 0.04 |
| Ethnicity, n (%) | | | | | | |
| White | 31 (22.6) | 146 (22.8) | 62 (20.1) | 0.97 | 0.54 | 0.34 |
| Pakistani | 74 (54.0) | 382 (59.6) | 183 (59.2) | 0.23 | 0.30 | 0.91 |
| Others | 32 (23.4) | 149 (22.0) | 95 (27.9) | 0.73 | 0.31 | 0.04 |
| **Frequency of comorbidities, n (%)** | | | | | | |
| Allergic disease | 136 (99.3) | 465 (68.7) | 0 (0.0) | <0.01 | <0.01 | <0.01 |
| Appendicitis | 3 (2.2) | 17 (2.5) | 5 (1.5) | 0.83 | 0.58 | 0.28 |
| Arthritis | 0 (0.0) | 1 (0.1) | 0 (0.0) | 0.65 | N/A | 0.48 |
| Celiac disease | 3 (2.2) | 1 (0.1) | 2 (0.6) | 0.05 | 0.44 | 0.78 |
| Constipation | 6 (4.4) | 38 (5.6) | 10 (2.9) | 0.56 | 0.43 | 0.06 |
| FD | 0 (0.0) | 3 (0.4) | 0 (0.0) | 0.44 | N/A | 0.22 |
| GORD | 11 (8.0) | 40 (5.9) | 13 (3.8) | 0.35 | 0.06 | 0.16 |
| IBD | 1 (0.7) | 0 (0.0) | 1 (0.3) | 0.03 | 0.51 | 0.16 |
| Migraine | 1 (0.7) | 12 (1.8) | 3 (0.9) | 0.38 | 0.87 | 0.27 |
| EDS, JHS | 1 (0.7) | 3 (0.4) | 0 (0.0) | 0.66 | 0.12 | 0.22 |
| Autism | 0 (0.0) | 0 (0.0) | 2 (0.6) | N/A | 0.37 | 0.05 |
| Intellectual disability | 0 (0.0) | 0 (0.0) | 0 (0.0) | N/A | N/A | N/A |
| Mother's abdominal pain | 0 (0.0) | 474 (70.0) | 157 (46.2) | <0.01 | <0.01 | <0.01 |
| Mother's allergic diseases | 137 (100.0) | 271 (40.0) | 81 (23.8) | <0.01 | <0.01 | <0.01 |
| Mother's appendicitis | 2 (1.5) | 7 (1.0) | 4 (1.2) | 0.66 | 0.80 | 0.84 |
| Mother's arthritis | 10 (7.3) | 77 (11.4) | 14 (4.1) | 0.53 | 0.50 | <0.01 |
| Mother's Celiac disease | 0 (0.0) | 9 (1.3) | 4 (1.2) | 0.18 | 0.20 | 0.84 |
| Mother's chronic fatigue syndrome | 1 (0.7) | 2 (0.3) | 0 (0.0) | 0.44 | 0.12 | 0.32 |
| Mother's chronic muscle pain | 1 (0.7) | 24 (3.5) | 4 (1.2) | 0.08 | 0.67 | 0.03 |
| Mother's constipation | 0 (0.0) | 8 (1.2) | 2 (0.6) | 0.56 | 0.37 | 0.37 |
| Mother's depressive disorder, bipolar disorder | 28 (20.4) | 252 (37.2) | 0 (0.0) | <0.01 | <0.01 | <0.01 |
| Mother's FD | 1 (0.7) | 27 (4.0) | 4 (1.2) | 0.21 | 1.00 | 0.04 |
| Mother's GORD | 14 (10.2) | 128 (18.9) | 14 (4.1) | 0.04 | 0.05 | <0.01 |
| Mother's IBD | 0 (0.0) | 5 (0.7) | 5 (1.5) | 0.31 | 0.15 | 0.26 |
| Mother's IBS | 12 (8.8) | 86 (12.7) | 24 (7.1) | 0.75 | 1.00 | 0.02 |
| Mother's migraine | 15 (10.9) | 230 (34.0) | 20 (5.9) | <0.01 | 0.24 | <0.01 |
| Mother's EDS, JHS | 0 (0.0) | 3 (0.4) | 1 (0.3) | 0.44 | 0.53 | 0.72 |
| Mother's obsessive-compulsive disorder | 2 (1.5) | 5 (0.7) | 3 (0.9) | 0.40 | 0.58 | 0.81 |
| Mother's intellectual disability | 0 (0.0) | 0 (0.0) | 0 (0.0) | N/A | N/A | N/A |
| Mother's schizophrenia | 1 (0.7) | 0 (0.0) | 1 (0.3) | 0.03 | 0.51 | 0.16 |

FD, functional dyspepsia; GORD, gastro-oesophageal reflux disease; IBD, inflammatory bowel disease; IBS, irritable bowel syndrome; EDS, Ehlers-Danlos syndrome; JHS, joint hypermobility syndrome.

of GI disorders was low in all phenotypes and showed few significant differences between the phenotypes. The clinical characteristics of each phenotype are summarised below.

**Phenotype 1.** Most children in Phenotype 1 had allergic diseases (99.3%), and all mothers had allergic diseases (100%), which were significantly higher than in Phenotypes 2 and 3 ($p<0.01$). In short, this phenotype was characterized as 'AP with allergic predisposition', suggesting relevance of allergic mechanisms in AP development in children.

**Phenotype 2.** In this phenotype, the frequency of allergic diseases is relatively high at 68.7%, but other comorbidities in the children were uncommon. In contrast, maternal comorbidities showed the highest frequencies in this phenotype: AP (70.0%), allergic diseases (40.0%), depressive and/or bipolar disorder (37.2%), GORD (18.9%), and migraine (34.0%). We termed Phenotype 2 as 'AP with mothers' comorbidities'.

**Phenotype 3.** The frequency of mother's AP (46.2%) was the second highest in Phenotype 3. However, the frequencies of other comorbidities, including children's allergic diseases and other maternal comorbidities were uncommon. This phenotype was termed 'AP with the least comorbidities'.

**Impact of allergic diseases and maternal comorbidities on paediatric AP.** We further investigated the effect of allergic diseases and maternal comorbidity burdens on the frequency of paediatric AP. As the number of paediatric allergic diseases increased, the frequency of AP in children increased commensurately (Fig 3A). Specifically, 17.6% of children with ≥ 3 allergic diseases experienced AP, which was significantly more frequent than those with 0–2 allergic diseases ($p<0.01$). Similarly, as the number of maternal comorbidities increased, the frequency of AP in children also increased (Fig 3B). In cases where mothers had ≥ 3 comorbidities, the frequency of paediatric AP was 25.6%, which was significantly higher than in cases with 0–2 comorbidities ($p<0.01$). Furthermore, when comparing the high-risk children (those who met the criteria of having both ≥ 3 allergic diseases and ≥ 3 maternal comorbidities) to the others (control), the high-risk group had a significantly higher frequency of AP ($p<0.01$) (Fig 3C).

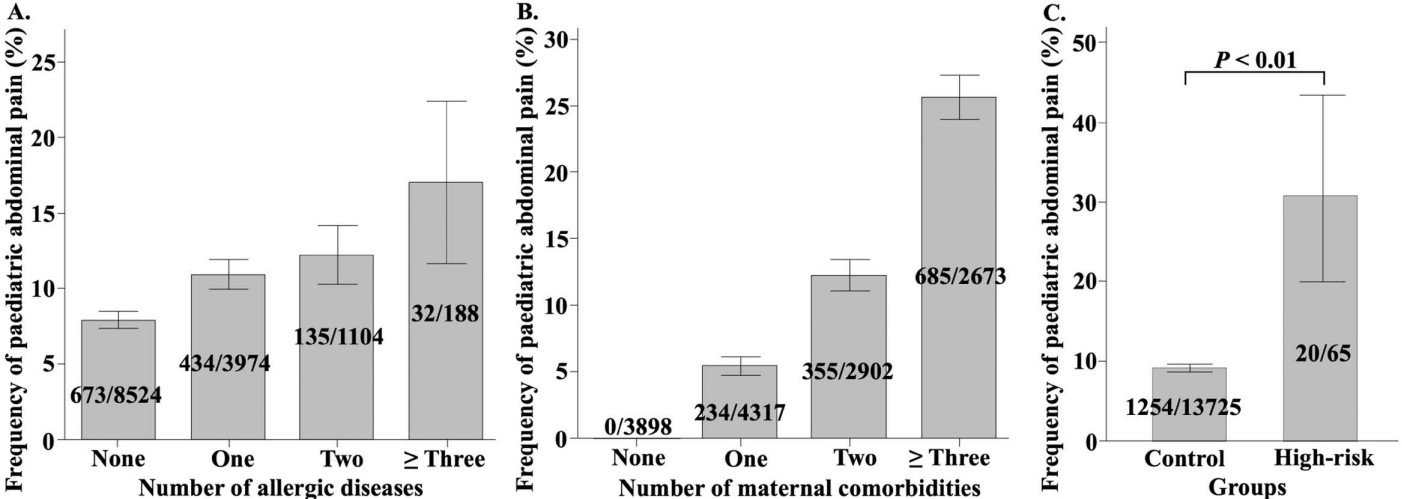

**Fig 3. The frequency of paediatric abdominal pain depending on the number of allergic diseases and the number of maternal comorbidities.** **A–B.** The bar graphs illustrate the relationship between the frequency of paediatric abdominal pain (AP) and the number of allergic diseases in children, as well as the number of comorbidities in their mothers. Error bars in the figure represent the 95% confidence intervals (CI) for the frequency of paediatric AP. **C.** The high-risk group is defined as children with both ≥ 3 allergic diseases and ≥ 3 maternal comorbidities while the remaining children are regarded as the control group. The frequency of AP is compared between the high-risk and control groups.

### 3.3. Study 3. Development of ML predictive models for paediatric AP

*Summary: The best predictive model for paediatric AP was CatBoost (AUC 0.67), which enabled moderate risk stratification. Ethnicity, especially Pakistani ethnicity, was the most important predictor, and maternal AP was the most influential maternal comorbidity.*

Finally, we developed predictive models for paediatric AP. After evaluating several algorithms, CatBoost yielded the highest AUC of 0.67 (95% CI: 0.63–0.71) and was therefore selected for subsequent analyses (Fig 4A and S6 Table). The accuracy, sensitivity, specificity, PPV, and NPV of the CatBoost-based model were 0.62 (95% CI: 0.58–0.65), 0.68 (95% CI: 0.63–0.73), 0.55 (95% CI: 0.50–0.61), 0.62 (95% CI: 0.57–0.66), and 0.62 (95% CI: 0.57–0.68), respectively (Table 4).

SHAP analysis indicated that ethnicity had a greater influence on the prediction than comorbidities, with White children being less likely and Pakistani children more likely to develop AP (Fig 4C and 4D). Maternal AP emerged as the most influential maternal comorbidity, while allergic diseases were the strongest predictor among paediatric comorbidities.

When stratified by the probability estimated by the CatBoost-based model, the observed rates of AP were 18.9%, 44.8%, 60.6%, and 100.0% in the <40%, 40–50%, 50–60%, and >60% probability groups, respectively (Fig 4B and Table 5). Compared to the lowest-probability group (<40%), the odds of AP were significantly higher in the 40–50% group (OR 3.49, 95% CI: 1.94–6.26, p<0.01), 50–60% group (OR 6.61, 95% CI: 3.76–11.6, *p*<0.01), and >60% group (OR could not be calculated, but *p*<0.01).

## 4. Discussion

The prevalence of paediatric AP in our study cohort was 9.2%. Several child and maternal factors, including paediatric ethnicity, GI and allergic diseases, as well as maternal AP, were associated with paediatric AP. ML-based clustering successfully identified three distinct phenotypes of paediatric AP, suggesting the presence of subgroups among children with AP. The characteristics of these phenotypes indicate that allergic diseases and maternal comorbidities are relevant to the development of AP in two of the three phenotypes. The predictive performance of the ML model was moderate when using the data from GP records. The most influential determinants of paediatric AP in the predictive model were child ethnicity, allergic predisposition, and maternal comorbidities, particularly maternal AP. Taken together, our findings suggest that both unsupervised and supervised ML approaches could uncover clinically relevant factors and may serve as useful tools for hypothesis generation in future studies.

In this study, maternal GI comorbidities, such as AP and GORD, were significantly associated with paediatric AP. This finding is consistent with previous research showing that parental factors are related to functional AP in children [8,10,18]. For example, children of mothers with IBS or chronic pain tended to experience more GI symptoms or functional AP [18,28]. This association may be explained by parental influence on children's pain experiences, whereby children may model their parents' pain behaviours, and parents may reinforce their children's pain complaints through solicitous responses [18,28–30]. Moreover, functional GI disorders, which are common causes of AP, are influenced by genetic factors [31], suggesting that the observed association may be partly due to shared genetic predisposition between mothers and children. Our findings underscore the important link between maternal factors and paediatric AP.

The ML clustering revealed three distinct phenotypes of paediatric AP. In all phenotypes, paediatric AP were not fully explained by GI diseases. Notably, Phenotype 1 (10.8%) was characterized by the presence of allergic diseases. Previous studies have reported that functional AP is multifactorial condition, influenced not only by parental and genetic factors but also by intestinal mechanisms such as low-grade inflammation [32], psychological aspects including anxiety, visceral hypersensitivity mediated by enteric and central nervous system modulation [33], and post-infectious dysbiosis leading to disruption of the brain–gut axis [34]. Although the role of allergic mechanisms in paediatric AP remains inconclusive, early-life allergic conditions—including eczema, rhinitis, hay fever, and asthma—have been associated with an increased risk of functional GI disorders, such as IBS, later in childhood [9,17,35]. Mast cells, which play a central role in visceral hypersensitivity [36], have recently been shown to undergo localized, food-induced activation in the intestines of patients

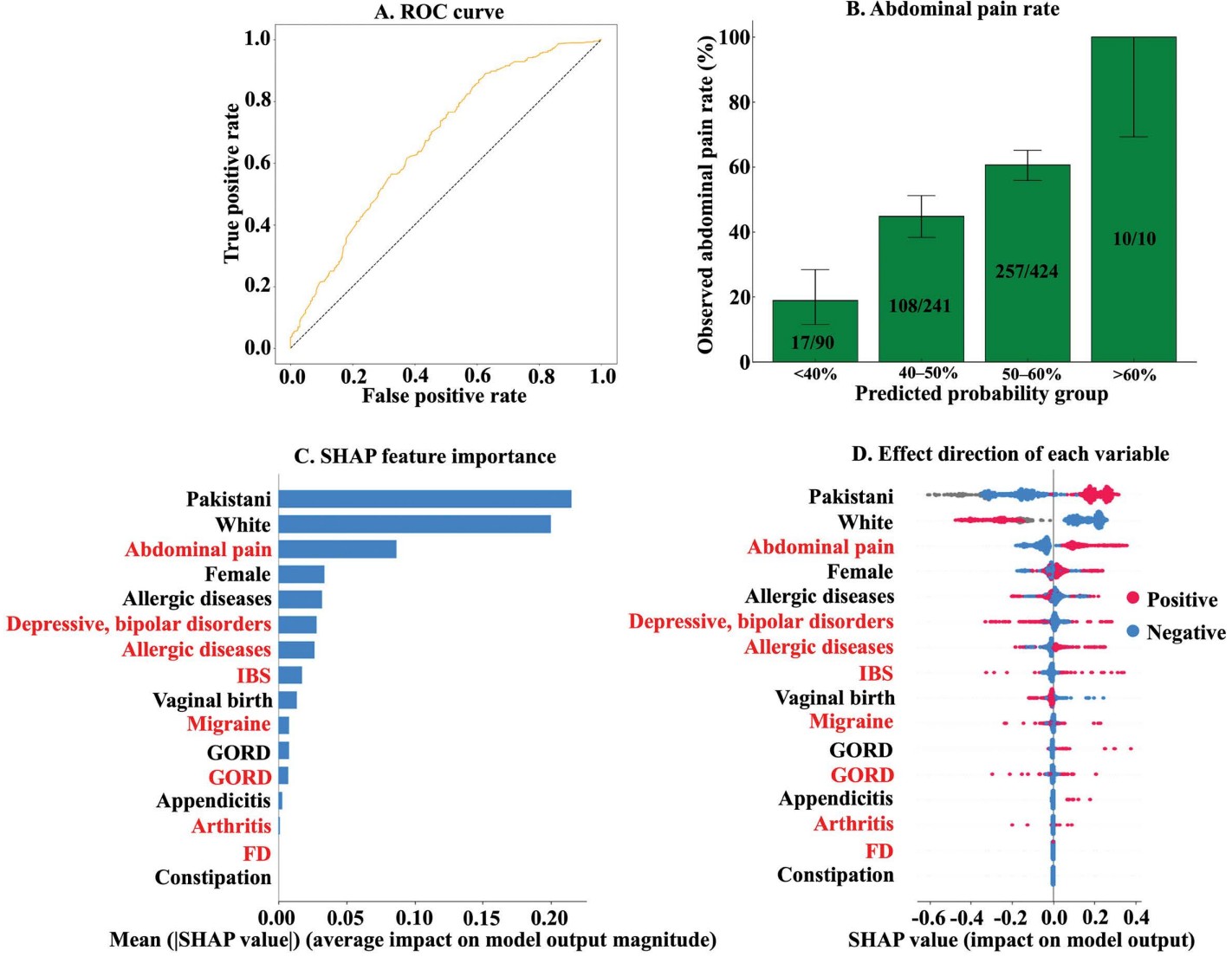

**Fig 4. Results of supervised machine learning predictive models for paediatric abdominal pain. A**. Receiver operating characteristic (ROC) curve of the CatBoost-based model. **B.** The y-axis represents the observed abdominal pain (AP) rate, while the x-axis indicates the predicted probability groups. Error bars represent the 95% confidence intervals of the observed AP rates. **C.** The mean Shapley Additive exPlanations (SHAP) value of each variable. Variables written in red represent maternal comorbidities, while those written in black represent variables related to the children themselves. **D.** The beeswarm plots show the SHAP feature importance and the direction of the effect of each variable on the model. The horizontal axis represents the SHAP value. Since all the variables used in the predictive models were binary, red and blue plots indicate positive or negative for each variable, respectively. For example, in the case of 'Pakistani', red plots tend to be distributed in the positive SHAP value and blue plots in the negative SHAP value. This indicates that Pakistani children are more likely to have AP.

with IBS [37]. Our findings, together with these reports, generate the hypothesis that gastrointestinal neuro-immune interactions may contribute to AP in children with allergic predispositions. Indeed, a recent study demonstrated that histamine-1 receptor antagonists effectively reduced AP intensity compared with placebo in patients with non-constipated IBS [38]. While these mechanisms require further validation in future studies, they may ultimately inform the development of targeted, allergy-focused interventions for pediatric AP.

**Table 4. Results of predictive models for paediatric abdominal pain.**

| Performance metrics | Outcomes |
|---|---|
| AUC (95% CI) | 0.67 (0.63–0.71) |
| Accuracy (95% CI) | 0.62 (0.58–0.65) |
| Sensitivity (95% CI) | 0.68 (0.63–0.73) |
| Specificity (95% CI) | 0.55 (0.50–0.61) |
| PPV (95% CI) | 0.62 (0.57–0.66) |
| NPV (95% CI) | 0.62 (0.57–0.68) |

Abbreviations: CI, confidence interval; PPV, positive predictive value; NPV, negative predictive value; AUC, area under the curve.

**Table 5. Risk stratification of paediatric abdominal pain based on the probability predicted by the CatBoost-based ML model.**

| AP probability by ML | Observed AP rate, % (n/N) | OR (95% CI) | p value |
|---|---|---|---|
| <40% | 18.9 (17/90) | ref | |
| 40–50% | 44.8 (108/241) | 3.49 (1.94–6.26) | <0.01 |
| 50–60% | 60.6 (257/424) | 6.61 (3.76–11.6) | <0.01 |
| >60% | 100.0 (10/10) | Inf | <0.01 |

Abbreviations: AP, abdominal pain; CI, confidence interval; ML, machine learning; OR, odds ratio.

In Phenotype 2 (53.1%), where maternal comorbidities were associated, aforementioned parental mechanisms may play a role in the development of paediatric AP [18,28–31]. In this subgroup, treatment strategies may need to consider not only interventions directed at the child but also parental factors, potentially including approaches such as cognitive behavioural therapy [39].

In contrast, in Phenotype 3 (26.7%), maternal AP was relatively common, occurring in 46.2% of cases. Therefore, some of the AP in children within Phenotype 3 may be explained by maternal influence. However, since the frequency of maternal comorbidities was much lower than in Phenotype 2, it is unlikely that all cases can be attributed to this factor. It is possible that the information available from GP records was insufficient to deeply phenotype children in Phenotype 3. Furthermore, 9.4% of children in this study were categorized as unclassified. With more detailed AP-related data, such as socioeconomic status, lifestyle, autonomic nervous system function, and gut microbiota composition, it may be possible to perform a deeper phenotypic analysis of children in Phenotype 3 and identify unclassified children as a distinct phenotype with specific characteristics [40–42].

The CatBoost-based model achieved a moderate AUC of 0.67 and identified the child's ethnicity as a key factor, consistent with its significance in the logistic regression model. The BiB cohort is unique in that it primarily includes children of White (29.3%) and Pakistani ethnicities (39.5%) [15]. Socioeconomic status varies by ethnicity within the cohort [43], and lower socioeconomic status is a known risk factor for pain conditions in children, including AP [44]. Furthermore, socioeconomic status is also associated with the prevalence of allergic disease, and previous studies have reported a higher prevalence of allergic diseases among children of Pakistani origin in the BiB cohort [43]. In the present dataset, the frequency of allergic diseases was significantly higher in children of Pakistani origin compared with other ethnic groups (45.1% vs. 38.9%, $p < 0.01$). Allergic diseases were the main characteristic of Phenotype 1 in our clustering analysis and were identified as a potential factor associated with AP. Thus, the significance of Pakistani origin in the predictive model may partly reflect the higher prevalence of allergic diseases in this group. Importantly, the use of a cohort with a high prevalence of allergic diseases may have influenced both the phenotyping and predictive modelling results, and caution is warranted

when generalizing these findings to the wider population. Nevertheless, this unique cohort setting may also be regarded as a strength, as it has enabled the generation of new research questions that warrant further investigation.

Notably, approximately 60% of Pakistani couples represented in this cohort are reported to have consanguineous marriages, with 37% being first-cousin unions [45]. Children born from consanguineous marriages have been shown to have higher mortality rates and an increased number of primary care appointments compared to those born to non-consanguineous couples, indicating a higher burden of health issues [46]. Such consanguinity may also help explain why being of Pakistani ethnicity was a significant risk factor for paediatric AP in our model.

While the ML predictive model enabled a certain degree of risk stratification according to predicted probabilities, the substantial number of AP cases observed even in lower (40–50%) and intermediate (50–60%) probability groups suggests that the model's discriminative power remains limited. This may be attributable to the fact that, although important associated factors such as allergic diseases and maternal AP were included as predictors in the ML model, variables directly reflecting the underlying pathophysiology of AP were not incorporated. Future models may benefit from incorporating variables that capture the underlying pathophysiology of AP, such as visceral hypersensitivity, altered gut-brain signalling, dysbiosis, and subclinical intestinal inflammation [47–49].

Our study is not without limitations. First, we were unable to clearly distinguish between chronic and acute AP due to the limited temporal detail in the available GP records. However, given the rarity of organic diseases that typically cause acute AP, the majority of cases were presumed to be chronic. While this limits the precision of phenotype classification, it reflects real-world clinical practice where such distinctions are not always made or documented clearly. Moreover, the identified risk factors–such as allergic predisposition– are likely relevant to both types of AP, as supported by previous literature [17,50]. Most of our findings are also consistent with prior studies, lending support to the credibility and reliability of the data [40]. Therefore, despite this limitation, our findings provide valuable insights into the broader mechanisms of paediatric AP and remain relevant for understanding its development. Second, at the GP level, the exact cause of AP might not always be identifiable. Therefore, the criteria used for each diagnosis and the exact causes of AP in each case are unknown and was beyond our control. However, functional AP is likely the predominant cause, as previous community-based studies of AP in children suggest that approximately 80% of cases are classified as functional or non-medically explained, whereas organic diseases account for only 5–10% of cases [42]. Despite its complexity, real-world data of this nature holds significant potential for generating valuable insights, particularly when analysed using advanced methods such as ML [51]. Finally, although we evaluated the performance of our predictive models in an out-of-sample test partition, we did not fully validate them using a different cohort of children, which would further maximize generalizability. Moreover, a different cohort would also be useful for verifying the reproducibility of clustering. As this was an exploratory study, further research is warranted to examine the identified factors in greater detail.

In conclusion, our exploratory study identified paediatric and maternal comorbidities as significant associated factors for paediatric AP. By applying data-driven clustering techniques, we delineated three distinct phenotypes of paediatric AP. These findings may lay the groundwork for future research aimed at clarifying the underlying pathophysiological mechanisms, validating these results in more diverse populations, and exploring potential genetic and environmental contributors. The predictive models developed in this study highlight the potential for early identification and intervention. Further studies incorporating more detailed data could refine phenotyping and predictive models, leading to more accurate predictions and paving the way for optimised, patient-personalised treatment strategies.

## Supporting information

**S1 Fig.  Workflow for developing predictive models.** The entire dataset was randomly partitioned into 70% for model training and 30% for out-of-sample testing. Using the training data, we trained predictive models. Grid-search with 10-fold cross-validation was employed to tune the hyperparameters of each model using the training dataset. The training dataset was divided into 10 folds. Nine folds were used as the training dataset, and the remaining one-fold was used as the

validation dataset. This process was repeated 10 times to optimize the hyperparameters of each predictive model. The performance of each model was evaluated using the test dataset.
(PDF)

**S1 Table. Investigated diseases and their corresponding systematized nomenclature of medicine clinical terms (SNOMED-CT) codes.**
(DOCX)

**S2 Table. Hyperparameters of UMAP and HDBSCAN.**
(DOCX)

**S3 Table. Frequency of variables used in machine learning predictive models.**
(DOCX)

**S4 Table. Characteristics of all children (N = 13,790).**
(DOCX)

**S5 Table. Non–significant results of logistic regression analysis for the diagnosis of paediatric abdominal pain.**
(DOCX)

**S6 Table. Results of predictive models for paediatric abdominal pain using various machine learning algorithms.**
(DOCX)

## Author contributions

**Conceptualization:** Kazuya Takahashi, Humayoon Satti, Qasim Aziz.

**Data curation:** Humayoon Satti.

**Formal analysis:** Kazuya Takahashi, Huma Shehwana.

**Funding acquisition:** Kazuya Takahashi.

**Supervision:** James K. Ruffle, John A Williams, Animesh Acharjee, Georgios V Gkoutos, Qasim Aziz.

**Writing – original draft:** Kazuya Takahashi, Georgios V Gkoutos.

**Writing – review & editing:** Michalina Lubiatowska, Huma Shehwana, James K. Ruffle, John A Williams, Animesh Acharjee, Shuji Terai, Humayoon Satti, Qasim Aziz.

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
