## [Decision Letter · Decision Letter 0]

18 Sep 2025

PLOS ONE

Dear Dr. Takahashi,

Thank you for submitting your manuscript to PLOS ONE. After careful consideration, we feel that it has merit but does not fully meet PLOS ONE’s publication criteria as it currently stands. Therefore, we invite you to submit a revised version of the manuscript that addresses the points raised during the review process.

ACADEMIC EDITOR COMMENTS: 

We look forward to receiving your revised manuscript.

Kind regards,

Hany Mahmoud Abo-Haded, MD

Academic Editor

PLOS ONE

 [This study was supported by JA Niigata Kouseiren Grant (Niigata University School of Medicine) and JSPS Grants-in-Aid for Scientific Research (grant number: 22K16013).]. 

Additional Editor Comments:

Reviewer #1:

Reviewer #2:

Reviewers' comments:

Reviewer's Responses to Questions

**Comments to the Author**

1. Is the manuscript technically sound, and do the data support the conclusions?

Reviewer #1: Partly

Reviewer #2: Yes

2. Has the statistical analysis been performed appropriately and rigorously?

Reviewer #1: Yes

Reviewer #2: Yes

3. Have the authors made all data underlying the findings in their manuscript fully available?

Reviewer #1: Yes

Reviewer #2: Yes

4. Is the manuscript presented in an intelligible fashion and written in standard English?

Reviewer #1: Yes

Reviewer #2: Yes

Reviewer #1: The manuscript was a great read, introducing AI and Abdominal Pain (AP). AI and medical associations are the wave of the future but still needs perfecting. In agreeance with AI facilitating a deep dive into the human diagnosis. It can be difficult in choosing the supervised and unsupervised learning as they are both beneficial in human exploration. Unsupervised learning needs the human expertise with the outcomes. There are area of this manuscript that need details on technique and methods used, as AI is in its introductory phase. The association is a bit farfetched or novel.

Major Revision/Recommendations:

1. Supervised vs unsupervised learning: From a clinical aspect, unsupervised was the correct step but why not complete both methods. You identified in your manuscript that unsupervised learning was implemented due to exploratory features, I agree. This should be explained or made clear. Also, supervised could be very beneficial, in predicting future outcomes.

2. Illustrations should be added using cluster analysis, hierarchical structure of the identified clusters, etc. Emerging topics should be visual, so that your audience receives your point of view.

3. The cluster were mainly focused on allergic disease which is not a common in abdominal pain. I would explain this in detail, mentioning this finding from beginning to end. Are you able to explain the algorithm or coding. The result of allergic disease associated with abdominal pain is a bit farfetched with their being more common associations.

4. The significance of the population being mostly Pakistan and White could be multifactorial (environmental, etc.)

5. The phenotypes seemed unmatched with actual societal groups with abdominal pain or maybe novel new finding.

6. If AI is main objective, this is great but if using the association to come to a conclusion more work is needed. If this is a manuscript on allergic disease, AI and maternal comorbidities, great read but to associate with Abdominal Pain is a bit farfetched.

7. Your audience needs to be taught ad convinced that AI is appropriate for science and will generate conclusive results.

Reviewer #2: The study by Kazuya Takahashi et al. explores pediatric abdominal pain (AP) using machine learning (ML) to identify phenotypes and predict risk factors. Analyzing data from 13,790 children in the Born in Bradford cohort, the researchers identified three AP phenotypes: allergic predisposition, maternal comorbidities, and minimal comorbidities. Allergic diseases and maternal health issues significantly increased the frequency of AP, with 17.6% of children having ≥3 allergic diseases and 25.6% of children with ≥3 maternal comorbidities experiencing AP. A supervised ML model achieved moderate predictive performance (AUC 0.67), highlighting ethnicity, pediatric allergic diseases, and maternal comorbidities as key factors. Risk stratification showed AP rates ranging from 18.9% (<40% probability) to 100% (>60% probability). These findings emphasize the role of genetic, environmental, and maternal influences in the development of AP, offering insights for future research and personalized interventions.

The study question is relevant to clinical practice, and the authors conducted a thorough literature review. The analysis of data from 13,790 children provides robust statistical power and generalizability. The major findings are clearly presented, and the research objectives are effectively addressed. The results support the conclusions, and the manuscript is well-written.

However, the manuscript could be improved by addressing the following issues:

1. The authors should discuss in greater detail the gap in existing knowledge regarding AP phenotypes and predictive factors to better justify the need for the study.

2. A brief overview of the methods (e.g., ML clustering and predictive modeling) would help readers understand how the study addresses the research gap.

3. Summarizing the most important results at the beginning of each subsection in the Results section would help readers quickly grasp the main takeaways.

4. The Discussion section could be strengthened by incorporating more data from previous studies to contextualize the findings and highlight how this study advances knowledge on pediatric abdominal pain.

5. The authors should provide deeper insights into the clinical implications of the identified phenotypes, including how they could inform personalized treatment strategies.

6. More specific recommendations for future studies would be beneficial, such as incorporating pathophysiological data, validating findings in diverse populations, and exploring genetic factors.

**Do you want your identity to be public for this peer review?** For information about this choice, including consent withdrawal, please see our Privacy Policy

Reviewer #1: **Yes: ** Nicole Y Fatheree

Reviewer #2: No

---

## [Author Response · Author response to Decision Letter 1]

14 Oct 2025

13th October 2025

Emily Chenette

Editor-In-Chief

PLOS ONE

Dear Editor,

Re: An Exploratory Machine Learning Study on Paediatric Abdominal Pain Phenotyping and Prediction (PONE-D-25-34689)

Thank you for your decision dated 18th September 2025, enclosing the reviewers' comments. We have carefully reviewed the comments and have revised the manuscript accordingly. Our responses are given in a point-by-point manner below. Modifications to the manuscript are written in red.

We hope the revised manuscript is now suitable for publication and look forward to hearing from you in due course.

Yours sincerely,

Kazuya Takahashi

Centre for Neuroscience, Surgery and Trauma,

Wingate Institute of Neurogastroenterology, Blizard Institute,

Barts and the London School of Medicine and Dentistry,

Queen Mary University of London

Email: kazuya911@med.niigata-u.ac.jp

Comments from the academic editor

Please complete the explanation of both learning methods (the supervised and the unsupervised )

Response: Thank you for the valuable feedback and the opportunity to further clarify our methodology. We have expanded the description of both supervised and unsupervised machine learning approaches to provide a clearer understanding of their principles and applications in our study (page 9, line 17 to page 13, line 8). We hope that the revised explanation enhances the readers’ comprehension of our methods.

Comments from Reviewer 1

The manuscript was a great read, introducing AI and Abdominal Pain (AP). AI and medical associations are the wave of the future but still needs perfecting. In agreeance with AI facilitating a deep dive into the human diagnosis. It can be difficult in choosing the supervised and unsupervised learning as they are both beneficial in human exploration. Unsupervised learning needs the human expertise with the outcomes. There are area of this manuscript that need details on technique and methods used, as AI is in its introductory phase. The association is a bit farfetched or novel. Major Revision/Recommendations:

1. Supervised vs unsupervised learning: From a clinical aspect, unsupervised was the correct step but why not complete both methods. You identified in your manuscript that unsupervised learning was implemented due to exploratory features, I agree. This should be explained or made clear. Also, supervised could be very beneficial, in predicting future outcomes.

Response: Thank you for providing us with the opportunity to elaborate on our manuscript. We have clarified the rationale for adopting unsupervised ML clustering in the Methods section (page 10, lines 2–6), emphasizing that this approach was chosen because the study was exploratory and aimed to identify previously unrecognized phenotypes without pre-imposed labels.

As the reviewer points out, supervised learning is indeed beneficial for predicting future outcomes. We have already implemented a supervised predictive model in this study, and we have now revised the Methods section to make this clearer and to highlight its role in predicting abdominal pain (AP) occurrence. (page 11, line11–16)

2. Illustrations should be added using cluster analysis, hierarchical structure of the identified clusters, etc. Emerging topics should be visual, so that your audience receives your point of view.

Response: We sincerely appreciate this constructive suggestion. As recommended, we have added Condense Tree to illustrate the hierarchical structure of the identified phenotypes (Figure 2B) and plots color-coded by paediatric allergic diseases and representative maternal comorbidities (Figure 2C).

3. The cluster were mainly focused on allergic disease which is not a common in abdominal pain. I would explain this in detail, mentioning this finding from beginning to end. Are you able to explain the algorithm or coding. The result of allergic disease associated with abdominal pain is a bit farfetched with their being more common associations.

Response: Thank you for raising this important point regarding the our clustering result. We agree that the clustering procedure should be described in detail, and we have now added the parameters of UMAP and HDBSCAN in S2 Table for reproducibility.

We acknowledge that the finding of allergic diseases as a major focus is a novel finding in AP and therefore we have provided a more detailed description of our methods as requested and also in the discussion section as described below.

While allergic mechanisms may not represent the main pathophysiology of paediatric AP, previous reports have suggested their involvement, including in irritable bowel syndrome (page 21, line 15 to page 22, line 16). Thus, considering allergic mechanisms as a potential contributor to AP is not without precedent.

In our study, the inclusion of a unique cohort with a high prevalence of allergic diseases, particularly among children of Pakistani origin, may have contributed to this finding (page 23, line 17 to page 24, line 16). We believe it is possible that allergic mechanisms play a role in a subgroup of paediatric AP, and this possibility warrants further investigation. In this sense, we consider our study valuable as a foundation for future research.

4. The significance of the population being mostly Pakistan and White could be multifactorial (environmental, etc.)

Response: We appreciate the reviewer’s insightful comment. We agree that the ethnic composition of the cohort may have influenced the observed results. In particular, the BiB cohort includes a large proportion of participants of Pakistani origin, a population known to have a higher prevalence of consanguinity, which may contribute to unique genetic characteristics. In addition, cultural and dietary habits may differ from those of the other major group, White participants. Furthermore, as noted earlier, the prevalence of allergic diseases is significantly higher among individuals of Pakistani origin compared with other ethnic groups. These factors have been acknowledged and discussed in the Discussion section (page 23, line 15 to page 24, line 16).

5) The phenotypes seemed unmatched with actual societal groups with abdominal pain or maybe novel new finding.

Response: We agree that the identified phenotypes may not align with conventional societal groupings of AP. As noted in our response to Comment 3, we consider this result as potentially novel, generating new research questions linking allergy to paediatric AP and further research to understand the underlying mechanisms is now warranted. We have clarified this point more explicitly in the Discussion section (page 21, line 15 to page 22, line 16).

6) If AI is main objective, this is great but if using the association to come to a conclusion more work is needed. If this is a manuscript on allergic disease, AI and maternal comorbidities, great read but to associate with Abdominal Pain is a bit farfetched.

Response: As stated in our previous responses, this is an exploratory study using AI-based clustering, and the findings are not intended to establish definitive causal relationships between allergic diseases, other factors, and AP. We agree that drawing definitive conclusions would require studies with different designs and replication in independent cohorts. Nevertheless, we believe that our study has value in generating novel hypotheses and in highlighting research questions that warrant further investigation in future research. We have added further clarification regarding the future value of our research in our conclusion section on page 27, line 5–14.

7) Your audience needs to be taught ad convinced that AI is appropriate for science and will generate conclusive results.

Response: We appreciate this important comment. To help readers better understand the appropriateness of AI in this context, we have expanded the Methods section to provide a clearer description of the AI approach used (page 9, line 17 to page 13, line 8). In addition, we explicitly state that this approach is aimed at generating new hypothesis and research questions to facilitate for future research on this important topic (page 27, line 6–10).

Comments from Reviewer 2

The study by Kazuya Takahashi et al. explores pediatric abdominal pain (AP) using machine learning (ML) to identify phenotypes and predict risk factors. Analyzing data from 13,790 children in the Born in Bradford cohort, the researchers identified three AP phenotypes: allergic predisposition, maternal comorbidities, and minimal comorbidities. Allergic diseases and maternal health issues significantly increased the frequency of AP, with 17.6% of children having ≥3 allergic diseases and 25.6% of children with ≥3 maternal comorbidities experiencing AP. A supervised ML model achieved moderate predictive performance (AUC 0.67), highlighting ethnicity, pediatric allergic diseases, and maternal comorbidities as key factors. Risk stratification showed AP rates ranging from 18.9% (<40% probability) to 100% (>60% probability). These findings emphasize the role of genetic, environmental, and maternal influences in the development of AP, offering insights for future research and personalized interventions.

The study question is relevant to clinical practice, and the authors conducted a thorough literature review. The analysis of data from 13,790 children provides robust statistical power and generalizability. The major findings are clearly presented, and the research objectives are effectively addressed. The results support the conclusions, and the manuscript is well-written.

1) The authors should discuss in greater detail the gap in existing knowledge regarding AP phenotypes and predictive factors to better justify the need for the study.

Response: Thank you for providing us with the opportunity to elaborate on our manuscript. We have expanded the Discussion section to provide a more detailed description of the existing knowledge gap regarding the causes of AP, thereby better justifying the need for this study. In addition, as also noted by Reviewer 1, we have provided a more detailed discussion of the observed association between allergy and AP (page 21, line 15 to page 22, line 16).

2) A brief overview of the methods (e.g., ML clustering and predictive modeling) would help readers understand how the study addresses the research gap.

Response: As also noted by Reviewer 1, we have expanded the Methods section to include a brief description of the ML clustering and predictive modeling approaches used in this study (page 9, line 17 to page 13, line 8). We believe that these additions make it clearer why AI was employed and how the study addresses the identified research gap.

3. Summarizing the most important results at the beginning of each subsection in the Results section would help readers quickly grasp the main takeaways.

Response: Thank you for this helpful suggestion. We have revised the Results section to include a summary of the key findings at the beginning of each subsection, as recommended. We believe this change improves the readability of the section and allows readers to more easily grasp the main takeaways.

4. The Discussion section could be strengthened by incorporating more data from previous studies to contextualize the findings and highlight how this study advances knowledge on pediatric abdominal pain.

Response: Thank you for pointing out this issue. We have addressed this point in the Discussion by considering previous reports alongside our own findings and by outlining the hypothesis that emerges from this study (page 21, line 15 to page 24, line 16).

5. The authors should provide deeper insights into the clinical implications of the identified phenotypes, including how they could inform personalized treatment strategies.

Response: We have addressed it in the Discussion by describing potential personalized treatment strategies that could be considered in the future for Phenotype 1 (page 22, line 10–16) and Phenotype 2 (page 22, line 18 to page 23, line 3).

6. More specific recommendations for future studies would be beneficial, such as incorporating pathophysiological data, validating findings in diverse populations, and exploring genetic factors.

Response: We appreciate the reviewer’s suggestion. We have incorporated this point into the conclusion, noting the importance of clarifying the underlying pathophysiological mechanisms, validating these results in more diverse populations, and exploring potential genetic and environmental contributors (page 27 line 7–10).

---

## [Decision Letter · Decision Letter 1]

23 Oct 2025

An exploratory machine learning study on paediatric abdominal pain phenotyping and prediction

PONE-D-25-34689R1

Dear Dr. Takahashi,

We’re pleased to inform you that your manuscript has been judged scientifically suitable for publication and will be formally accepted for publication once it meets all outstanding technical requirements.

Kind regards,

Hany Mahmoud Abo-Haded, MD

Academic Editor

PLOS ONE

Additional Editor Comments (optional):

Reviewers' comments:

Reviewer's Responses to Questions

**Comments to the Author**

Reviewer #1: All comments have been addressed

Reviewer #2: All comments have been addressed

2. Is the manuscript technically sound, and do the data support the conclusions?

Reviewer #1: Yes

Reviewer #2: Yes

3. Has the statistical analysis been performed appropriately and rigorously?

Reviewer #1: N/A

Reviewer #2: Yes

4. Have the authors made all data underlying the findings in their manuscript fully available?

Reviewer #1: Yes

Reviewer #2: Yes

5. Is the manuscript presented in an intelligible fashion and written in standard English?

Reviewer #1: Yes

Reviewer #2: Yes

Reviewer #1: I appreciate your response to my review, and it reads much easier now for a focused academic reader and also a lay person not in the field. Thanks you for all your work with helping to bring AI to Healthcare. All the best.

Reviewer #2: The authors addressed my comments and incorporated my suggestions. In my opinion, the manuscript is ready for publication.

**Do you want your identity to be public for this peer review?** For information about this choice, including consent withdrawal, please see our Privacy Policy

Reviewer #1: **Yes: ** Nicole Y Fatheree

Reviewer #2: **Yes: ** Rajmohan Dharmaraj, MD

---

## [Editor Report · Acceptance letter]

PONE-D-25-34689R1

PLOS ONE

Dear Dr. Takahashi,

I'm pleased to inform you that your manuscript has been deemed suitable for publication in PLOS ONE. Congratulations! Your manuscript is now being handed over to our production team.

Kind regards,

on behalf of

Professor Hany Mahmoud Abo-Haded

Academic Editor

PLOS ONE